# Genomic Analyses of Wild and Cultivated Bacanora Agave (*Agave angustifolia* var. *pacifica*) Reveal Inbreeding, Few Signs of Cultivation History and Shallow Population Structure

**DOI:** 10.3390/plants11111426

**Published:** 2022-05-27

**Authors:** Anastasia Klimova, Karen Y. Ruiz Mondragón, Francisco Molina Freaner, Erika Aguirre-Planter, Luis E. Eguiarte

**Affiliations:** 1Departamento de Ecología Evolutiva, Instituto de Ecología, Universidad Nacional Autónoma de México, Ciudad Universitaria, Circuito Exterior s/n Annex to the Botanical Garden, Mexico City 04510, Mexico; karen.mondragon91@gmail.com (K.Y.R.M.); eaguirre@ecologia.unam.mx (E.A.-P.); 2Departamento de Ecología de la Biodiversidad, Instituto de Ecología, Universidad Nacional Autónoma de México Hermosillo, Sonora 83250, Mexico; freaner@unam.mx

**Keywords:** gene flow, genomic resources, genomic variation, genotyping by sequencing (GBS), single nucleotide polymorphisms, spatial structure

## Abstract

Due to the recent increase in demand for agave-based beverages, many wild agave populations have experienced rapid decline and fragmentation, whereas cultivated plants are now managed at monocultural plantations, in some cases involving clonal propagation. We examined the relative effect of migration, genetic drift, natural selection and human activities on the genetic repertoire of *Agave angustifolia* var. *pacifica*, an agave used for bacanora (an alcoholic spirit similar to tequila) production in northwestern Mexico. We sampled 34 wild and cultivated sites and used over eleven thousand genome-wide SNPs. We found shallow genetic structure among wild samples, although we detected differentiation between coastal and inland sites. Surprisingly, no differentiation was found between cultivated and wild populations. Moreover, we detected moderate inbreeding (*F*_IS_ ~ 0.13) and similar levels of genomic diversity in wild and cultivated agaves. Nevertheless, the cultivated plants had almost no private alleles and presented evidence of clonality. The overall low genetic structure in *A. angustifolia* var. *pacifica* is apparently the result of high dispersibility promoted by pollinators and the possibility of clonal reproduction. Incipient cultivation history and reliance on wild seeds and plants are probably responsible for the observed patterns of high genetic connectivity and considerable diversity in cultivated samples.

## 1. Introduction

Plant populations are not arbitrary assemblages of genotypes but are structured in space and time [1]. The major evolutionary forces—such as gene flow, genetic drift and different selection regimes—are responsible for the observed genetic structure of plant species and populations [2]. Moreover, plant populations can also be profoundly affected by human activities, such as habitat modification, domestication, direct extraction and introduction of invasive species [3]. Determining the relative importance of these forces may be extremely complicated, as several factors may be at work simultaneously [1]. Nevertheless, an understanding of the population genetic structure of a species can yield critical information for conservation, management and general understanding of plant evolution [1,4]. 

Migration among plant populations is affected by life history traits (e.g., mating system), pollinators and seed dispersal mechanisms, and by the continuity of geographical distribution [5,6,7]. Some pollinators, such as nectar feeding bats, can travel long distances, thereby reducing the structuring of genetic variation among pollinated populations [8,9,10], whereas other pollinators, such as bees, usually travel short distances [11,12], which would favor mating between closely spaced individuals [13]. Similarly, with seed dispersers, short-distance vectors (e.g., gravity) move seeds only a few meters, thereby limiting migration, whereas long-distance vectors (e.g., migratory birds or bats) move seeds much further, thereby facilitating migration and reducing genetic structure [10,13,14].

Another important factor affecting plant population genetic structure is spatial distribution [1,15]. Plant species can occur in uniform, clumped, or random distributions, or, for instance, along an elevation gradient; and the ranges of species may vary in size from widespread to very narrow and endemic. It is thought that plant species with a large or patchy distribution will exhibit stronger genetic structure than those with a small or continuous distribution, because it is more difficult to move pollen and seeds across larger spatial distances [16,17]. Moreover, species distributed across wide heterogenous environments may be exposed to differential selection regimes or to local adaptation [18]. In this case, the successful occupation of a wide area may be explained by the occurrence of many specialized genotypes, rather than by the existence of a single universal genotype. Therefore, divergent selection among these populations inhabiting ecologically different environments would create a barrier to gene flow, thereby promoting genetic differentiation [19,20].

Genetic structure in plants is also affected by human use and cultivation [21,22]. Usually, the most serious impacts on the effective population size (*N_e_*) and connectivity among plant populations comes from direct extraction and lumbering [21,23]. On the other hand, many plant species that have been historically cultivated may be located outside their natural range, and genetic isolation from source populations may lead to increased genetic structuring. Cultivation of a limited number of individuals may cause a founder effect, resulting in genetic drift, lower genetic diversity and increased differentiation [24]. Artificial selection for desirable traits in the process of domestication may also decrease genetic variation and further increase genetic structure [25]. Otherwise, cultivation may reduce genetic structure if genotypes are obtained from multiple source populations, or if gene flow occurs between cultivated and local wild populations [26,27]. Given the above, determining and explaining population genetic structure of plant species, although very important, may be challenging.

In this regard, species of *Agave* L. (Asparagaceae) may be excellent models to investigate the relative importance of natural and/or anthropogenetic causes that affect population structure and diversity in plants. *Agave* is a species rich genus of perennial monocots found primarily in the arid regions of Mexico and the southwestern United States [28,29]. *Agaves* are keystone species that disproportionately affect their ecological communities relative to their abundance by producing a large inflorescence with copious amounts of nectar and pollen [30,31,32]. On the other hand, agaves have a long history of interaction with humans and they have been utilized historically by native populations in North America as a source of fiber, food, beverages and medicine [31,33]. The production of spirits from *Agave* has a long history, and nowadays it has become the primary use of agave [29,31]. Over fifty species of *Agave* L. are used for beverage production [29,34]; yet, the majority of the utilized species are not cultivated but rather extracted from natural populations [29,35]. Nevertheless, a growing number of species are now managed [33] with different levels of intensity, from planting wild seeds in backyards to extensive monocultural plantations of laboratory produced clones with no flowering allowed [36]. In the last decades, the demand for tequila like beverages has grown dramatically; for example, since 2014 the production of mezcal has increased by almost 500% [37]. This trend has not only increased pressure on natural populations but has also introduced industrial level techniques into plant production and gathering [38]. 

One of the most important *Agave* spp. utilized in Mexico is *Agave angustifolia*. *A. angustifolia* is also the species with the widest geographic distribution within the genera; it is found from Sonora and Tamaulipas in Mexico to Costa Rica [31]. *A. angustifolia* is a diploid, apparently self-incompatible [39,40] species with a high rate of sexual reproduction [31]. It is pollinated by *Leptonycteris* bats, birds and bees [39]. Because of its wide distribution and unresolved phylogenetic relationships, *A. angustifolia* is sometimes considered to comprise several varieties [41]. In northwestern Mexico, particularly in the state of Sonora, the complex is represented by *A. angustifolia* var. *pacifica,* which is the focus of the present study. 

*Agave angustifolia* has a rich ethnobotanical history [31,42]. Currently *A. angustifolia* is the most important agave species used for mescal elaboration in Oaxaca, the state with the highest percentage (over 90%) of mescal production, and other places [41,43]. The scale of spirits production in Oaxaca is at an industrial level for both national and international markets. Therefore, thousands of hectares have been destined to agave cultivation. The selection of desired phenotypes has led to a system of monoculture cultivars (varieties) derived from *A. angustifolia*—“Espadín” [41]. In Sonora, on the other hand, *A. angustifolia* is used for production of bacanora [44], a less known beverage, popular mainly in northwestern Mexico. Although, cultivation of *A. angustifolia* var. *pacifica* is now often carried out in monocultural extensive areas, the majority of plantations are small, with important reliance on wild seeds and plants. 

The increase in mezcal production, and in particular, the accelerated extraction of wild individuals, constitutes an important environmental problem [45]. This issue is well exemplified by tequila expansion, which promoted rapid growth of plantations, massive destruction of natural vegetation, soil erosion, reliance on vegetative propagation, and prevention of cross-pollination and gene flow [38]. As a consequence, the genetic diversity of *A. tequilana* was drastically reduced [46,47], which led to an increased vulnerability to pathogens [48]. The available studies on other agave species indicate that there may be a considerable reduction in genetic diversity in cultivated individuals [47,49,50]. On the other hand, several studies have reported that owing to the incipient stage of domestication and recent formation of crop areas, many agave species still maintain relatively high genetic diversity. However, there is already a trend towards a decrease in genetic variation and increase in differentiation [47,51,52,53]. Apparently, the reduction of genetic diversity and increased differentiation are related to the intensity and time under management [54]. Given the above, it is of the utmost importance to apply state-of-the-art methodologies to evaluate natural and anthropogenic factors that may affect population structure and diversity of wild and cultivated agave species, and therefore, avoid losses of genetic variation and evolutionary potential. 

In this study, we characterized the patterns of genomic diversity at varying geographic scales, and investigated the relative importance of natural forces and human activities in shaping patterns of genetic structure of *A. angustifolia* var. *pacifica*, an agave used for bacanora (an alcoholic spirit similar to tequila and mezcal) production in the Sonora state in northwestern Mexico. We analyzed 96 individuals from wild and cultivated sites of *A. angustifolia* var. *pacifica,* genotyping them with over 11,000 SNP markers (Appendix A). Our aims were to (1) determine whether and how geographic distribution, life history traits (e.g., mating system, dispersal mechanisms, longevity) and human activities have affected patterns of genetic structure and diversity in this species; and (2) determine the extent to which we can detect genetic signatures of intensifying human management in the wild and cultivated populations. Due to the fact that *A. angustifolia* pollinators may potentially move genetic material across long distances, we hypothesized that a shallow population structure among sites and considerable genetic diversity would be observed in wild plants. We also expected that due to the patchier geographical distribution, history of cultivation and possible genetic drift, individuals under management would show less genetic diversity and stronger genetic structure relative to wild populations. Such an approach has the potential to aid conservation and management strategies because it can identify at-risk, low-diversity wild and cultivated sites that would benefit from restored gene flow within a broader geographic region.

## 2. Results

The genotyping by sequencing (GBS) on 96 *A. angustifolia* samples resulted in a total of 407,485,082 PE raw reads, with an average of 3,953,325 reads per sample (range 2,006,790–5,647,966). After filtering and adapters removal, the final data set resulted in an average of 2,630,790 high quality reads per sample (range 1,114,581–3,785,957). From this data, 638,704 variants were called using the *de novo* pipeline. After filtering, our final data set consisted of 95 *A. angustifolia* individuals and 11,619 SNPs with 2.8% of the data missing. Average sequencing depth for each individual and missingness on per site and per individual level are presented in Appendix A.

### 2.1. Population Structure 

Overall, populations of *A. angustifolia* showed shallow genetic differentiation, with the exception of some cultivated sites and one wild site (Figure 1, Appendix A). First of all, although different from zero, genetic differentiation between wild and cultivated plants was low. For example, the averaged paired *F*_ST_ was 0.005 (95% CI, 0.0046–0.0058). Similar results were found using Nei’s *D* index, as cultivated plants had negligible differentiation from their wild counterparts (0.006). We further carried out differentiation analyses at site level for cultivated and wild individuals and only for the wild sites (Appendix A). For the complete data set, we found that one wild and three cultivated sites presented comparatively high differentiation from the rest of the samples (*F*_ST_ ranged from 0.1 to 0.57). One of these sites corresponded to a plantation that had in vitro propagated plants (MocC3; personal communication with owner; [36])*,* another corresponded to samples from an intensively managed monocultured plantation and the last corresponded to a backyard plantation from a small village. Only one of the wild sites (CamW) was highly differentiated (*F*_ST_ > 0.35; Appendix A) from the rest of the wild samples. After excluding this site (CamW), the differentiation among wild sites was moderate (*F*_ST_ = 0.076, ranging from −0.01 to 0.22; Appendix A). We also found considerable differentiation between coastal sites (elevation lower than 10 m) and the rest of the samples (*F*_ST_ ranged from 0.09 to 0.22). When these coastal sites were excluded, the differentiation was low and homogenously distributed, ranging from *F*_ST_ −0.01 to 0.01 (Appendix A). 

To investigate the fine scale population structure between and within wild and cultivated *A. angustifolia*, we used an ADMIXTURE analysis. When all samples (cultivated and wild) were used, the cross-validation error estimates showed that model fit was optimized at *K* = 4 (Appendix A). Nevertheless, to better understand the genetic structure within the samples, we plotted the results from *K* = 3 to *K* = 5 (Figure 1). 

In general, ADMIXTURE analyses found shallow genetic structure within *A. angustifolia* var. *pacifica,* with only one wild site and several cultivated individuals being differentiated. Based on the complete data set, we were able to group individuals into the following clusters (corresponding to *K* = 5; Figure 1C): (i) wild individuals from one site (CamW) plus two cultivated individuals from the Moctezuma region (in yellow); (ii) three individuals from a backyard plantation of the Mazatan region (in dark blue); (iii) three individuals from an intensively managed plantation near Navojoa (in green); (iv) ten wild individuals from the coastal area near San Carlos and Navojoa (in purple); and (v) all the rest of the cultivated and wild samples. Some individuals from the last group (v) also presented mixed ancestry with samples from the coastal group (iv) (Figure 1A–C). 

A PCA showed patterns consistent with those described by the pairwise *F*_ST_ comparisons and ADMIXTURE clustering (Figure 2). 

After removing outlier samples (three wild individuals of the CamW population and tree samples from the backyard cultivated plants from the Mazatan region, Appendix A), and except for a handful of cultivated samples, all individuals clumped into one cloud, with no apparent differentiation between management type or site (Figure 2A). The cultivated samples that were separated from the main cluster were: (i) in vitro plants and (ii) plants from one of the intensively managed plantations near Navojoa city. When we focused only on the wild samples (Figure 2B), in the first eigenvector (which explained 5.2% of the variance) most individuals formed a cloud, with some separated individuals from the coastal site of the San Carlos region. The second eigenvector (3.7%) further separated coastal samples from the southernmost sites near Navojoa from the rest of the samples.

### 2.2. Spatial Genetics 

For the spatial analysis, we focused only on the wild samples, also excluding one sampling site (CamW) that presented higher differentiation. The cross-validation criterion performed with *TESS3R* did not exhibit a minimum value or a plateau (Appendix A), but there was a steady increase in statistical support at higher *K* values. However, after *K* = 5, biologically meaningful structure was lost. Therefore, we plotted the results of different values of *K* (from *K* = 2 to *K* = 4; Figure 3).

At *K* = 2 (Figure 3A), we found support for differentiation of two coastal southernmost sites. Further partitioning at *K* = 3 added the San Carlos region coastal samples as a separate group (Figure 3B). At *K* = 4, we observed differentiation of the central sites collected between Novillo and Bacanora villages that was not observed with any other analysis (Figure 3C). Spatial PCA analysis revealed highly significant global (*p* = 0.001) but non-significant (*p* = 0.89) local spatial structures (Appendix A), indicating signatures of among-site separations. A plot of lagged scores from the first two principal components suggested that the global structure was linked to the elevation and differences of the coastal environment (Appendix A), corroborating with the *TESS3R* results and pairwise *F*_ST_. Taken together, we interpret that our sampling area has at least three genetic clusters: (i) samples of two coastal sites from low elevation near Navojoa city; (ii) a group including three coastal sites of low elevation near San Carlos; and (iii) an inland group comprising the rest of the sites.

Genetic distance (*F*_ST_) between wild sampling sites increased with geographical distance (Mantel test = 0.41, *p* = 0.007); nevertheless, the relationship was not linear (Appendix A). Moreover, a plot of *F*_ST_ vs. elevation distance revealed a less strong but still significant relationship (Mantel test *r* = 0.19, *p* = 0.01; Appendix A). 

### 2.3. Genetic Diversity

The overall mean observed heterozygosity (*H_o_*) for all samples was 0.22 (SD 0.13). The mean *H_o_* among 53 wild agave individuals was lower (0.22, SD 0.12) than the expected heterozygosity (*H_e_* = 0.25, SD 0.13), indicating a deficit of heterozygous individuals. Almost identical results were obtained for the cultivated samples (*H_o_* = 0.22, SD 0.13 and *H_e_* = 0.25, SD 0.13). The difference between expected and observed heterozygosity was significant, even after Bonferroni correction (Bartlett’s test all samples K-squared = 84.965, *p*-value < 0.001; only wild samples K-squared = 36.168, *p*-value < 0.001). Individual based multilocus heterozygosity (MLH) was similar to the *H_o_* for both wild and cultivated plants (Appendix A). In addition, two frequency-weighted measures of individual heterozygosity—standardized multilocus heterozygosity (sMLH) and internal relatedness (IR)—showed no difference between wild and cultivated individuals (Appendix A). 

We found that wild and cultivated agave plants were moderately inbred according to the Fhat3 inbreeding index, with similar *f* averaging 0.13 (SD 0.01) for both wild and cultivated agaves. In wild individuals, *f* ranged from almost zero to as high as 0.37 (Appendix A), while in cultivated plants, inbreeding ranged from 0.09 to 0.24 (Appendix A). Similar results were obtained using Wright’s *Fis* (Appendix A). 

When samples were partitioned at the population/region level and by management type, we found a similar pattern of homogeneous distribution of heterozygosity and inbreeding (Appendix A). The lowest inbreeding (0.11) was found in wild populations of Navojoa and Mazatan, whereas the highest estimates (0.15) were found in two wild populations and one of cultivated plants (Appendix A). Nevertheless, after FDR correction, none of the comparisons were significant. Further partitioning of wild samples was carried out based on the spatial clustering of the samples recovered with *TESS3R* (Table 1). We found significant (*p* = 0.02) differences in MLH and *F_is_* between inland and coastal sites, with coastal sites presenting lower diversity and higher inbreeding (Table 1 and Appendix A). 

The mean relatedness was 0.007 (CI 0.002) (Figure 4), indicating a very low relationship among most sampled individuals. Nevertheless, we did find high (over 0.5) levels of relationship between some of the comparisons (12 pairs of individuals). These presumable clones were represented by plants sampled from intensively managed (NavC) and in vitro propagated (MocC) populations.

When the number of private alleles was compared between managed and wild plants, we found that wild individuals harbored a higher number of alleles that were not shared with cultivated plants (i.e., “private” alleles), 299 vs. 12.

## 3. Discussion

We analyzed patterns of genetic diversity and population structure of wild and cultivated *A. angustifolia* var. *pacifica* from northwestern Mexico. *Agave angustifolia* is a species with a wide distribution and a long history of human use that recently has escalated up to intensive management and heavy pressure on wild populations (Molina Freaner pers. observ.). Nevertheless, probably due to a combination of factors, including aspects of the species biology (e.g., long life cycle, monocarpic reproduction), pollination biology (involving bats and other highly mobile animals), the incipient cultivation history and particularities of bacanora production (a spirit less known in the market than tequila and mezcal, lower production and high reliance on wild individuals), we found low genetic structure, and homogeneously distributed levels of diversity in both wild and cultivated individuals. As for the wild samples, we found low but detectable spatial population genomic structure, apparently driven by the ecological characteristics of the habitat.

### 3.1. Population Structure in Wild and Cultivated Agaves

One of the aims of this study was to understand how natural forces, life history traits and cultivation practices may affect patterns of genetic structure of *A. angustifolia*. We found that wild populations of *A. angustifolia*, with a relatively continuous distribution in dry, mid-elevation areas throughout northwestern Mexico [42], do not present strong population genetic structure, which is not surprising, considering that most agaves have low genetic differentiation [49]. 

Overall, genetic structure is created by limited migration among populations, genetic drift and different selection regimes occurring across populations [2]. In *A. angustifolia* var. *pacifica*, shallow genetic structure is probably the result of high genetic connectivity, which homogenizes populations and reduces the effects of genetic drift and selection. Species of nectar-feeding bats in the genus *Leptonycteris* have been proposed as important pollinators of agaves in general, and of *A. angustifolia* in particular [39,55], Agave species pollinated by bats consistently present similarly low population genetic structure [49,50,52]. Interestingly, these bat species themselves have a wide distribution and general panmixia with low genetic differentiation among colonies [55,56,57].

Another important life history trait that may contribute to the observed lack of differentiation is the possibility of clonal reproduction and, as a consequence, enhanced longevity of genets [31,42,58]. In the harsh desert environment, temporal gaps between years with successful sexual recruitment can be highly variable. Therefore, plants can benefit from clonal reproduction, as clonality can enhance genet longevity, compensate for the partial loss of genets due to disturbance, reduce effective population size fluctuations and, thus, decrease the effects of genetic drift [31,42,59]. Moreover, the high longevity of clones may imply that even if the establishment of seeds dispersed over long distances is rare, and individuals arising from seeds dispersed over long distances are outnumbered in a local population, the likelihood of detection and sexual reproduction of such individuals is increased because they persist for longer periods of time [60]. 

Although we did not detect population genetic structure using traditional methods, we did find interesting spatial clustering of the samples. Thus, we detected that low elevation coastal sites (with an elevation of less than 50 m) are differentiated from the inland genetic cluster. Moreover, those sites were also differentiated from each other, a fact that may also explain non- linearity in Mantel tests. Coastal regions have unique environmental characteristics, such as high salinity, greater availability of atmospheric moisture, wind strength and tidal influence, therefore constituting distinct landscapes with unique abiotic and biotic composition [61,62]. We therefore hypothesize that different selection regimes occurring across populations in the coastal environment may be responsible for the observed differentiation. Further studies based on loci located in coding regions or focusing on transcriptome analysis would be crucial in determining genomic regions involved in promoting this low but significant differentiation [63]. 

The genetic differentiation between wild and cultivated agaves has been the focus of extensive research [29,49], with variable results. For example, some intensively cultivated varieties and species, such as *A. tequilana,* present considerable genetic differentiation from its wild relatives and among cultivars [54,64]. Basic explanations for the observed structure include reproductive isolation through asexual reproduction and high genetic drift. On the other hand, insignificant divergence related to management was observed in other agave species [49,53]. We found that, with the exception of a few managed sites, there is no genetic differentiation between wild and cultivated agaves in northwestern Mexico; in other words, the cultivated plants are basically a subset of the wild populations. The cultivation history of *A. angustifolia* var. *pacifica* is relatively recent and therefore, there may have been little time for differentiation.

Our findings expand the knowledge of genetic differentiation and diversification processes in agaves across coastal and inland areas of northwestern Mexico. Our results are concordant with previous suggestions [1,58] that multiple processes are likely influencing genetic structure in natural plant populations. We suggest that pollinator-mediated gene flow and clonal reproduction may serve as homogenization forces, whereas different selection regimes may be responsible for promoting local adaptation and intraspecific differentiation. Our results suggest that a complex mixture of features related to life history traits, environment and management shapes contemporary genetic differentiation patterns of *A. angustifolia* var. *pacifica*. 

### 3.2. Genetic Diversity 

The most widespread and concerning consequence of plant management and domestication is the erosion of genetic diversity [65,66]. During the initial phases of domestication and due to the effects of genetic bottleneck, genetic variation is usually reduced through decreasing effective population size and increasing inbreeding [25]. Then, additional loss of diversity is observed due to artificial selection through the removal of allelic variants of genes underlying traits that are undesirable for cultivation [67,68]. To date, genetic erosion and reduction in diversity of gene expression have been observed in many intensively cultivated plant species [69,70].

The case of agave is probably more complicated, with some cultivated varieties (e.g., *A. tequilana, A*. *salmiana* and *A. americana*) presenting low levels of diversity [46,54,71,72], whereas in other species levels of genetic variability are indistinguishable or even higher in cultivated individuals in comparison to their wild counterparts [49,53]. Here, we found that cultivated individuals of *A. angustifolia* in northwestern Mexico have not lost genomic diversity, nor did they present increased inbreeding in comparison to their wild counterparts. These findings are consistent with other studies reporting that traditional management of agave that involves gathering of wild specimens and seeds, instead of clonal propagation, allows the persistence of high genetic diversity in cultivated agaves [47,52,53]. On the other hand, in many traditional agave plantations, the diversity can be even greater than in wild individuals, due to the different origins of the germplasm introduced into cultivation and to non-strict selection [41,47]. This was also true in our study, since we found considerable genetic divergence (*F*_ST_ > 0.3) of some cultivated sites and individuals, particularly at some of the intensively managed plantations. Unfortunately, we were not able to determine the original geographic source of these samples, as probably they were moved from locations outside the sampling area. We suggest that future studies should focus on analysis of *A. angustifolia* throughout its whole distribution area and to establish a standard set of markers that would be useful to trace the origin of the cultivated individuals and to compare their diversity levels. 

The incipient domestication in agaves may be targeting particular genomic regions [68,73], and therefore, the putatively neutral markers used here may not be efficient enough to identify small differences between wild and cultivated plants. Future studies may consider markers located at coding regions or transcriptome analysis [74,75]. Additionally, recently developed pangenome analysis may offer exciting opportunities and high precision in identifying specific regions and genes that are being lost/gained during the process of domestication and management [76,77].

Importantly, although we found that heterozygosity and inbreeding estimates were similar between wild and cultivated individuals, wild samples harbored a larger number of unique (private) alleles (299 vs. 12), which represent an important gene pool of standing genetic variation not found in the cultivated plants used for bacanora production. This genetic resource could be relevant, for example, in future crop improvement programs. These differences indicate that the cultivated plants do not include the total genomic diversity of *A. angustifolia*, since they are just a small sample of the diversity found in wild agaves.

### 3.3. Inbreeding 

In contrast to the general expectation that the lowest levels of inbreeding in plants would occur in outcrossing species, a significant excess of homozygotes and moderate levels of inbreeding (average *f =* 0.13 for both wild and cultivated plants) were observed, and these occurred consistently among sites. Interestingly, in the *Agave* genus, relatively high levels of inbreeding (*F*_IS_) have also been reported in other studies [49]. For example, considerable inbreeding was reported in wild and cultivated samples of *A. maximiliana* (*F*_IS_ range from 0.002 to 0.32; [53]), in *A. americana*, *A*. *salmiana* and *A*. *mapisaga* (*F*_IS_ ranged from 0.19 to 0.72 [72]) and in *A. potatorum* (*F*_IS_ ranged 0.21–0.32; [78]). 

Explaining moderate levels of inbreeding coupled with the considerable polymorphism that have been found in many plants has not been an easy task [79]. One interesting hypothesis states that individual reproductive variation and temporal variation in flowering among individuals, coupled with the long life of plants, may split populations into sublines, with occasional mixing of sublines. Within sublines, inbreeding and genetic drift are increased, but genetic drift of the entire population is decreased. Therefore, the sublines maintain both high levels of polymorphism and explain the homozygote excess [80]. Moreover, Trame et al. [81] estimated that the inbred progeny of *Agave schottii* apparently do not experience strong negative selection and, as a consequence, inbred individuals continue to make up a substantial proportion of the population. Nevertheless, this hypothesis has not been tested in *A. angustifolia*, and further research focusing on optimal outcrossing distances and effects of inbreeding depression in this species would be helpful in elucidating how moderate levels of inbreeding and low population structure can be maintained [81,82]. 

### 3.4. Conservation and Management Implications 

The major concern of the current increase of mezcal and related beverage production is that in the eagerness for satisfying ever-growing demand, there could be an unprecedent negative impact on natural habitat and on wild agave populations [53,83]. Currently, it is difficult to evaluate *A. angustifolia’s* level of genetic diversity in a broader context of Agavoideae species, and it is unclear what levels of diversity and inbreeding should be expected. These challenges result from the general lack of studies that cover the whole distribution range of this or other agave species and a myriad of different molecular markers that have been used over the last 50 years [29,49].

Nevertheless, we discovered an alarming trend in some cultivated samples, particularly collected from the intensively managed plantations. Although at each plantation we sampled distant individuals, we found several samples with a high relatedness coefficient (over 0.5), indicating clonal propagation. These findings are in contrast with the fact that almost all wild individuals and samples from the traditionally managed plantations were genetically unrelated. Clonal propagation has several advantages for cultivation; for example, it is easier and faster than propagation from seeds, and it also ensures that favorable genotypes and mutations are passed on to the next generation [84]. Nevertheless, a long-term absence of sexual recombination under exclusive clonality would lead to erosion of genetic diversity and loss of evolutionary potential [85]. Without recombination, the mutational load increases, ultimately leading to lower fitness and decrease in agronomic performance [84,86]. We therefore argue that if the reliance on vegetative propagation were to increase, ultimately it may be detrimental to bacanora production [87]. 

Based on our findings, we argue that there is an urgent need to conduct systematic research focused on (i) delineating distribution limits of agave species used for spirits production; (ii) determining conservation and management units within each species; (iii) evaluating the impact of wild plant extraction by humans on species distribution and numbers; (iv) developing a set of genomic markers that would allow reproducibility and comparison among studies. Moreover, ancient and historical genetic data, such as those from specimens stored in herbarium collections and remains from archaeological sites, can add a temporal dimension to conservation of agaves by providing baseline levels of diversity [88].

## 4. Materials and Methods

### 4.1. Sample Collection

We analyzed 96 adult plants of *A. angustifolia* collected from 34 sites in the state of Sonora, Mexico in October 2021 (Figure 5 and Appendix A).

From those, 20 sites belonged to wild populations, 11 sites represented Bacanora plantations, and three sites were backyard plantations of *A. angustifolia*. More details on geographic coordinates, number of specimens collected at each site and management type can be found in Appendix A. Fresh material from the field and cultivated specimens were stored at −20 °C until DNA extraction. 

### 4.2. DNA Extraction, Library Preparation and Sequencing 

For DNA extraction, three individuals were chosen randomly from each sampling site to reach a total of 96 plants. Total genomic DNA was extracted from disrupted lyophilized with liquid nitrogen leaf tissue using a modified CTAB protocol [89,90]. In short, we finely grinded the leaves and added 600 µL of CTAB buffer. Samples at 10,000 rpm were centrifuged for 8 min. We removed the supernatant and resuspended the pellet in 600 µL of CTAB buffer. Then samples were incubated at 65 °C for 15 min and kept in ice for 15 min. We added 600 µL of chloroform:octanol 24:1 to each sample, which was vortexed and centrifuged at 13,000 rpm for 15 min. We collected the supernatant in a new tube with cold absolute ethanol and 1/10 of the volume of cold sodium acetate. The precipitation step was carried out at 4 °C overnight. After that, we centrifuged samples at 13,000 rpm for 10 min, removed the supernatant, added 1 mL of 70% ethanol and centrifuged at 13,000 rpm for 10 min. Finally, we removed the supernatant and re-suspended the pellet in water (For more details see Appendix A). 

Total genomic DNA was checked for degradation using a 1.5% agarose electrophoresis gel. The quantity of the DNA was determined using Qubit 3.0 fluorometer with a Qubit dsDNA broad-range kit. Sample standardization and library preparation were performed at the University of Wisconsin Biotechnology Center. Each DNA sample was digested using a combination of methylation-sensitive restriction enzymes (PstI/MspI). The choice of enzymes was based on a previous standardization for *Agave salmiana* and *Agave lechuguilla*. A unique barcode was ligated to each sample; after that, all samples were combined and sequenced at the University of Wisconsin Biotechnology Center using NovaSeq 2 × 150. 

### 4.3. Bioinformatics Analysis 

The raw data were filtered by removing adapters and low-quality bases using TRIMMOMATIC [91]. For the initial filtering, the following parameters were used: LEADING:25, TRAILING:25, SLIDINGWINDOW:4:20 and MINLEN:60. After first quality filtering, reads in FASTQ format were demultiplexed, filtered and GBS loci were *de novo* assembled using ipyrad v. 0.9.79 [92] with parameters recommended for paired-end GBS data (https://ipyrad.readthedocs.io/, accessed on 20 February 2022). As no published genome of any *Agave* species is yet available, we used a *de novo* assembly strategy. We set the level of sequence similarity for clustering at 90% after comparing the number of recovered single nucleotide polymorphisms (SNPs) using values between 85–95% (see Appendix A). The final data filtering was performed using VCFtools v.0.1.15 [93]. Only loci with a mean minimum depth (across individuals) of over 14 and maximum 2 alleles with no InDels (no insertions or deletions) were kept. Additionally, we set a minor allele frequency at 0.05, to reduce the possibility of removing true rare alleles that are important in elucidating fine-scale structure [94]. We excluded sites on the basis of the proportion of missing data, keeping sites with no more than 10% missing data (--max-missing 0.9). One individual was removed from the dataset due to a high percentage of missing data (i.e., greater than 50%), which was likely caused by poor DNA quality. We then filtered out the variants that significantly deviated from HWE (*p* ≤ 0.05 after Bonferonni correction). Finally, to ameliorate the confounding effects of linkage disequilibrium (LD), we eliminated markers within the specified distance from one another using --thin argument (--thin 250) as implemented in VCFtools. The final VCF file was then produced for all downstream analyses.

### 4.4. Patterns of Genetic Structure in Wild and Cultivated Individuals 

Several complementary approaches were used to test for patterns of genetic structure between wild and cultivated agave plants and among sites. We estimated genetic distance between wild and cultivated samples, and among all pairs of sites using the *StAMPP* package [95]. Clustering among samples without *a priori* grouping was inferred using the method implemented in ADMIXTURE v.1.23 [96,97]. Admixture analysis was run using 10,000 bootstraps; the number of clusters was set from 1 to 12 (*K*), with ten replicates for each *K* value. The support for different values of *K* was assessed according to the likelihood distribution (i.e., lowest cross-validation error), as well as by visual inspection of the co-ancestry values for each individual. We also used the R package *SNPRelate* to perform a principal component analysis (PCA) of *A. angustifolia* individuals based on the genetic covariance matrix calculated from the genotypes [98]. We ran two separate PCAs for *A. angustifolia* samples: (i) including both wild and cultivated plants, and (ii) including only wild individuals. 

### 4.5. Spatial Genetics 

To explore the role of geography in the population genetic structure of *A. angustifolia*, we used the spatially explicit clustering program *TESS3R* [99], which determines genetic variation in natural populations considering simultaneously genetic and geographic data. We tested *K* = 1–10 possible genetic groups with 20 replicates of each *K* and kept the most supported model (i.e., “best” based on cross-entropy scores) within each of the 20 replicates. Map locations were colored by the resulted dominant ancestry cluster, with transparency reflecting the percent ancestry of that cluster and the largest value assigned as opaque. 

While classic PCA is appropriate for detecting obvious genetic structure [100], it does not take spatial information into account and may therefore miss cryptic spatially arranged genetic structure. In order to confirm and visualize the genetic structure detected by spatial clustering in *TESS3R*, we conducted a spatial principal components analysis (sPCA) [101,102]. This procedure is a spatially explicit multivariate analysis that allows focusing on the part of the genetic variation that is spatially structured by optimizing not only the genetic variance between samples, but also their spatial autocorrelation. sPCA was conducted only for the wild samples using the package *adegenet* [101]. Tests for global and local genetic structure according to the definition given by Thioulouse et al. [103] were made using 1000 permutations. The visualization of detected structure was performed by plotting the samples according to their geographic coordinates, and coloring them according to their respective scores along the first and second sPCA components. 

Finally, we employed a Mantel test using the *ade4* package to estimate the correlation between the genetic distance, geographical and altitudinal distance matrices [102]. The test uses Pearson’s regression coefficient between distance matrices and 10,000 randomizations. The statistics and the test’s *p* value correspond to the proportion of times the randomized regression coefficient is equal or greater than the observed one. We calculated the pairwise genetic distance matrices among sites using *F*_ST_ and pairwise geographical distance matrices using the Geographic Distance Calculator [104]. 

### 4.6. Genetic Diversity

Expected heterozygosity (*H_e_*) and observed heterozygosity (*H_o_*) were calculated for the full dataset, and separately for each management type using the R packages *adegenet* and *hierfstat* [101,105]. We also calculated multilocus heterozygosity for each individual, which is defined as the total number of heterozygous loci in an individual divided by the number of loci evaluated in the focal individual [106]. Further, two frequency-weighted measures of individual heterozygosity—standardized multilocus heterozygosity (sMLH) and internal relatedness (IR) were quantified using the R packages *Rhh* and *inbreedR* [106,107]. We used plink 1.9 [108] to calculate the inbreeding index Fhat3 [109,110] and Wright’s *F*_IS_ statistics. Following Marshall et al. [111], we designated inbreeding coefficients (*f*) of zero as ‘none’, below 0.125 as ‘low’, 0.125 ≤ *f* < 0.25 as ‘moderate’, and *f* ≥ 0.25 as ‘high’. Wilcox tests were then used to test for significant differences in the diversity indices between management types, sites and populations identified with spatial analysis in *TESS3R* using the R package *stats* [112]. We also estimated the relatedness coefficient between each pair of individuals using TrioML methodology as implemented in COANCESTRY v. 1.0.1.10 [113,114]. For relatedness analysis, we used 100 bootstraps and accounted for inbreeding using 100 reference individuals. Finally, we estimated the number of private alleles in wild and in cultivated plants using the *poppr* package [115].

## 5. Conclusions

The current study aimed to dissect the genetic architecture of bacanora agave (*A. angustifolia* var. *pacifica)* from northwestern Mexico using SNP markers. We focused on wild and cultivated sites composed by individuals of the morphologically predominant morphotypes. Nevertheless, even this criterion did not help us in sampling morphologically similar but genetically different plants at one wild site (CamW) and several cultivated sites. Unfortunately, due to the considerable gap in agave phylogenetic and population genetic studies, we could not trace the geographic original source of these genotypes. Besides these outlier cultivated samples, we did not detect genetic differentiation between wild and cultivated individuals, neither did we find loss of genetic diversity in plants under management, suggesting that seeds of wild individuals are readily used for cultivation. Interestingly, the data analysis of wild samples revealed a partition of the samples into three groups—two coastal groups and one inland cluster. Nevertheless, being aware of our experimental limitations, at this moment we are unable to distinguish between local adaptation and neutral divergence; we therefore may only hypothesize reasons for such a separation.

## Figures and Tables

**Figure 1 plants-11-01426-f001:**
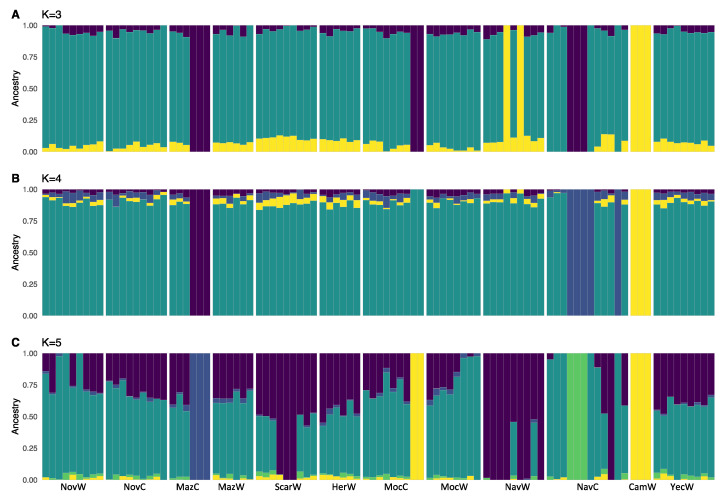
Clustering solutions obtained with 95 wild and cultivated individuals of *Agave angustifolia* var. *pacifica* from the state of Sonora, Mexico. Values of *K* from 3 to 5 are shown. In each plot (**A**–**C**), each cluster is represented by a different color, and each individual is represented by a vertical line divided into *K* colored segments with heights proportional to genotype memberships in the clusters. Thin white lines separate individuals from different regions and management types as coded in Appendix A (e.g., NovW corresponds to the Novillo region and wild samples, whereas MazC corresponds to the Mazatan region and cultivated samples).

**Figure 2 plants-11-01426-f002:**
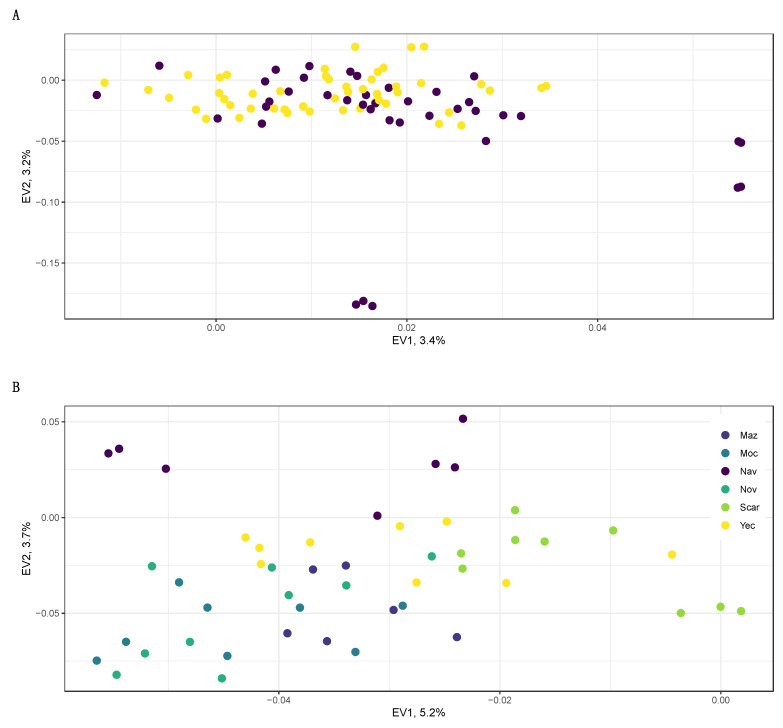
Relationships among (**A**) *A. angustifolia* var. *pacifica* from the state of Sonora, Mexico, wild (yellow) and cultivated (purple) samples as represented by principal component analysis (PCA) using 11,619 genome-wide SNPs and excluding 6 outlier samples. Panel (**B**) PCA representing relationships among wild samples of *A. angustifolia* var. *pacifica*. Colors corresponds to the major geographic regions detailed in Appendix A.

**Figure 3 plants-11-01426-f003:**
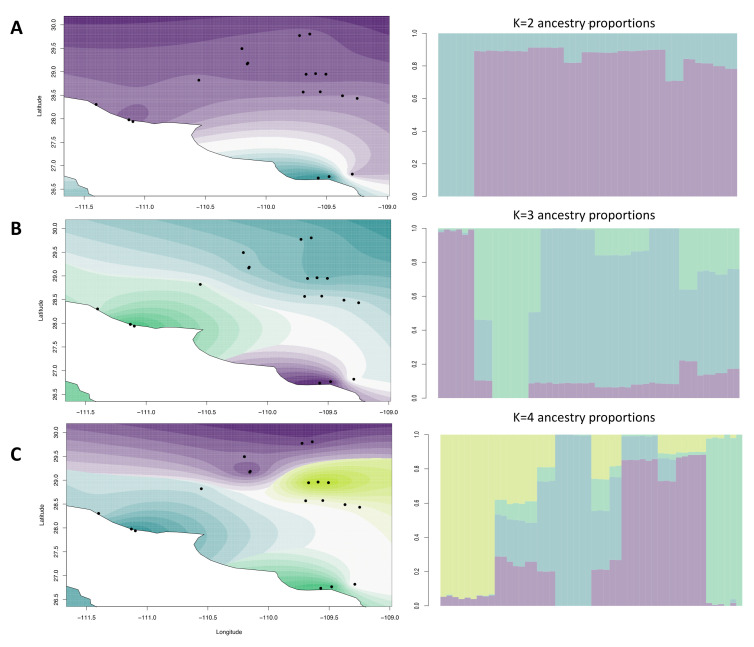
Interpolated ancestry proportions from TESS3R, demonstrating the geographic distribution of biologically meaningful genetic clusters (*K*) from 2 to 4 in *Agave angustifolia* var. *pacifica* from the state of Sonora, Mexico. Locations on each map are colored by the resulted dominant ancestry cluster, with transparency reflecting the percent ancestry of that cluster, with the largest value assigned as opaque. Each plot (**A**–**C**) corresponds to the respective values of *K* from 2 to 4.

**Figure 4 plants-11-01426-f004:**
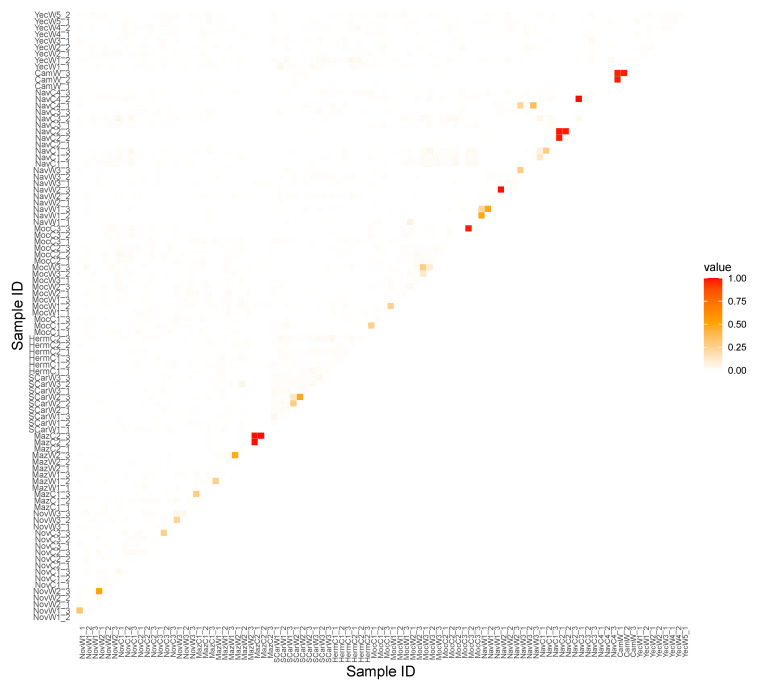
The overview diagram of relatedness coefficient as estimated with TrioML methodology among 95 *Agave angustifolia* var. *pacifica* individuals from the state of Sonora, Mexico. The degree of relatedness is represented by colors, from white indicating no relationship, to red representing a clonal relationship. Samples are coded according to the Appendix A.

**Figure 5 plants-11-01426-f005:**
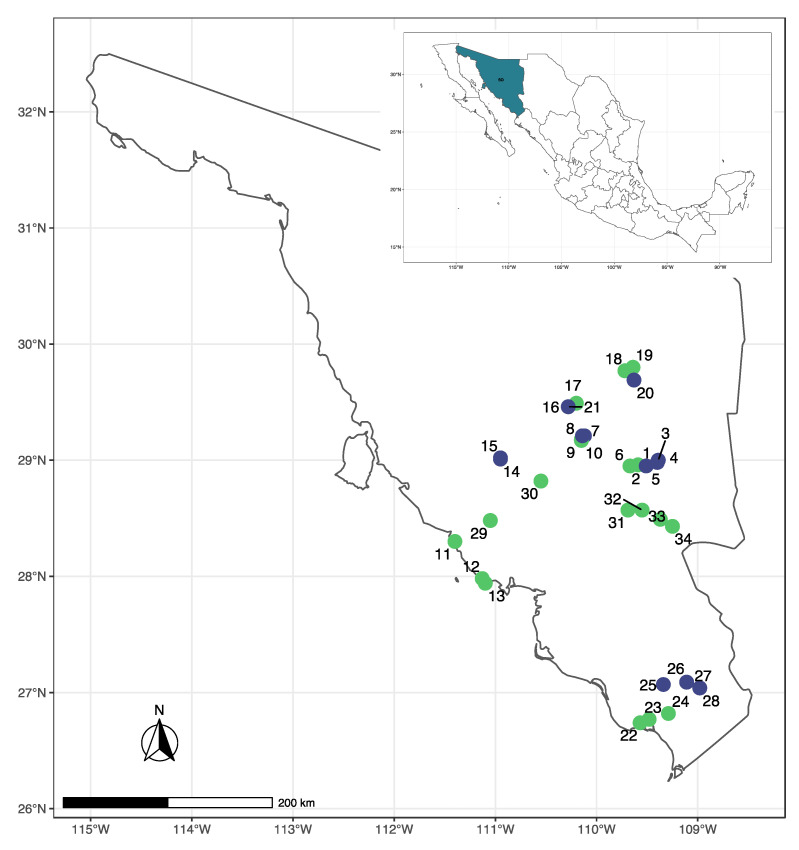
Map showing sampling sites of wild (green) and cultivated (blue) *Agave angustifolia* var. *pacifica* from the state of Sonora, Mexico. The full names of each site and sample sizes are given in Appendix A.

**Table 1 plants-11-01426-t001:** Diversity estimates and standard deviation (in parentheses) as estimated for wild and cultivated *Agave angustifolia* var. *pacifica* in the state of Sonora, Mexico. Wild populations were determined based on spatial analysis in TESS3R. One highly divergent wild site (Cam) is excluded from this table. *N*—number of individuals; sMLH—standardized multilocus heterozygosity; MLH—multilocus heterozygosity; IR—internal relatedness; *Fis*—Wright’s inbreeding index; Fhat3—inbreeding index.

Diversity Index	Wild Populations			Full Dataset Wild	Cultivated
	South Coastal	Central Coastal	Inland		
*N*	6	9	35	50	42
sMLH	1.00 (0.063)	0.98 (0.011)	1.00 (0.018)	0.99 (0.014)	1.00 (0.03)
MLH	0.22 (0.013)	0.22 (0.002)	0.22 (0.004)	0.22 (0.003)	0.22 (0.002)
IR	0.05 (0.048)	0.06 (0.007)	0.06 (0.014)	0.05 (0.011)	0.05 (0.02)
Fis	0.13 (0.054)	0.14 (0.009)	0.13 (0.016)	0.13 (0.012)	0.13 (0.03)
Fhat3	0.11 (0.047)	0.13 (0.001)	0.13 (0.017)	0.13 (0.012)	0.13 (0.02)

## Data Availability

The raw vcf file and additional stats file have been deposited in the ZENODO database with DOI: 10.5281/zenodo.6438381.

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
