# Peer review of "Genomic Analyses of Wild and Cultivated Bacanora Agave (Agave angustifolia var. pacifica) Reveal Inbreeding, Few Signs of Cultivation History and Shallow Population Structure"

_plants, 2022, doi:10.3390/plants11111426_

Round 1

Reviewer 1 Report

The study analyses the genetic structure of cultivated and wild populations of Agave angustifolia var pacifica from North west Mexico which is used in the production of bacanora spirit. The analysis is based on a set of >11000 SNP markers generated by a gbs approach. The rationale for the research is very well presented, the methods are sound and well described, and the conclusions are well argued. The language and writing of the article are of a high standard, making it easy to read and to follow the argumentation. I have found only a few very minor things that could be improved:

line 139 ...with 2.8% of the missing data...

should read ...with 2.8% of the data missing ...

line 479 ...Total genomic DNA was extracted from disrupted lyoph-ilized with liquid nitrogen leaf tissue using a modified CTAB protocol [91,92]...

the procedure is not clear to me. May be one or two sentences would be enough, as the protocol is probably only detailed in the cited PhD thesis (in Spanish) which is not easily accessible (ie citation [92]).

Author Response

Comments and Suggestions for Authors

The study analyses the genetic structure of cultivated and wild populations of Agave angustifolia var pacifica from North west Mexico which is used in the production of bacanora spirit. The analysis is based on a set of >11000 SNP markers generated by a gbs approach. The rationale for the research is very well presented, the methods are sound and well described, and the conclusions are well argued. The language and writing of the article are of a high standard, making it easy to read and to follow the argumentation. I have found only a few very minor things that could be improved:

 >>>Response 1.1. We are very grateful to the reviewer for positive feedback.

line 139 ...with 2.8% of the missing data...

should read ...with 2.8% of the data missing ...

>>>Response 1.2. Done. We changed the phrase as suggested by the reviewer.

line 479 ...Total genomic DNA was extracted from disrupted lyoph-ilized with liquid nitrogen leaf tissue using a modified CTAB protocol [91,92].

the procedure is not clear to me. May be one or two sentences would be enough, as the protocol is probably only detailed in the cited PhD thesis (in Spanish) which is not easily accessible (ie citation [92]).

 >>>Response 1.3. We apologize for this omission. We expanded “Methodology” section and included additional sentences detailing the steps of the extraction protocol. Lines 478-487. We also included detailed extraction protocol into Supplemental material.

Reviewer 2 Report

A good paper on a minor crop with increasing use value. 

If you cold present a better introduction to the crop evolution and distribution globally, where the plants belong geographically and where it was first cultivated, it would make sense, and then draw the line to the subspecies and to the current (and previous) cultivation in Mexico. As I understand the crop is domesticated in this area, which adds value to your conclusions. this could also fit into line 19 onward in the abstract.

Some minor issues:

Line 134, explain GBS first time you mention it. Look for similar shortcomings elsewhere also. 

Fig 4, axis details becomes messy to me.

Supplementary material: perhaps you could shorten the text here?

I see your ZENODO page is sill under embargo, but see it is there, so it should be fine, but be sure it is available upon publishing of the paper. 

Extensive and correct reference list. 

Thumbs up from here. The paper is good, well written and with a good methodology and discussion, Overall I have no problems here!

Author Response

A good paper on a minor crop with increasing use value. 

 >>>Response 2.1. We are grateful to the reviewer for the positive feedback.

If you could present a better introduction to the crop evolution and distribution globally, where the plants belong geographically and where it was first cultivated, it would make sense, and then draw the line to the subspecies and to the current (and previous) cultivation in Mexico. As I understand the crop is domesticated in this area, which adds value to your conclusions. this could also fit into line 19 onward in the abstract.

>>>Response 2.2. As suggested by the reviewer we now expanded Introduction section, including additional information regarding A. angustifolia distribution and cultivation. Lines 97-115

Some minor issues:

Line 134, explain GBS first time you mention it. Look for similar shortcomings elsewhere also. 

>>>Response 2.3. Thanks. We now checked the manuscript looking for unexplained abbreviations. We also explained the meaning of GBS in lines 29 and 154.

Fig 4, axis details become messy to me.

>>>Response 2.4. We improved the figure. It was a bit complicated to fit all the IDs. We changed the font to improve the axis.

Supplementary material: perhaps you could shorten the text here?

>>>Response 2.5. This part is a requisite from the journal, where we have to include titles of all the Supplementary Tables and Figures.

I see your ZENODO page is still under embargo, but see it is there, so it should be fine, but be sure it is available upon publishing of the paper. 

>>>Response 2.6. Yes, the data is still under embargo and it would be available once the manuscript is accepted.

Extensive and correct reference list. 

Thumbs up from here. The paper is good, well written and with a good methodology and discussion, Overall I have no problems here!

>>>Response 2.7. We are grateful for the positive appraisal